# Differentiating Post–Digital Nannying Autism Syndrome from Autism Spectrum Disorders in Young Children: A Comparative Cross-Sectional Study

**DOI:** 10.3390/jcm11226786

**Published:** 2022-11-16

**Authors:** Hamid Reza Pouretemad, Saeid Sadeghi, Reza Shervin Badv, Serge Brand

**Affiliations:** 1Institute for Cognitive and Brain Sciences, Shahid Beheshti University, Tehran 19839-69411, Iran; 2Center of Excellence in Cognitive Neuropsychology, Shahid Beheshti University, Tehran 19839-69411, Iran; 3Department of Pediatrics, Children’s Medical Center, Pediatrics Center of Excellence, Tehran University of Medical Sciences, Tehran 14166-34793, Iran; 4Center of Affective, Stress and Sleep Disorders (ZASS), Psychiatric Clinics (UPK), University of Basel, 4002 Basel, Switzerland; 5Division of Sport Science and Psychosocial Health, Department of Sport, Faculty of Medicine, Exercise and Health, University of Basel, 4002 Basel, Switzerland; 6Sleep Disorders Research Center, Kermanshah University of Medical Sciences, Kermanshah 67158-47141, Iran; 7Substance Abuse Prevention Research Center, Kermanshah University of Medical Sciences, Kermanshah 67158-47141, Iran; 8Department of Psychiatric, School of Medicine, Shahid Beheshti University of Medical Sciences, Tehran 14166-34793, Iran

**Keywords:** early excessive screen-time, ASD, PDNAS, lifestyle

## Abstract

Excessive exposure of young children to digital devices has increased in recent years. Much research has shown that early excessive screentime is associated with autistic-like symptoms. This study aimed to differentiate children with Post–Digital Nannying Autism Syndrome (PDNAS) from children with autism spectrum disorders (ASD) and typically developing children (TDC), both behaviorally and cognitively. This study is comparative and cross-sectional and included three groups of children. The first group consisted of 15 young children with subthreshold autism symptoms. They had not received a formal diagnosis of ASD and had been exposed to digital devices for more than half of their waking time. The second group consisted of 15 young children with ASD, and the third group consisted of 15 young TDC. A lifestyle checklist, a modified checklist for autism in toddlers (M-CHAT), a behavioral flexibility rating scale-revised (BFRS-R), the Gilliam autism rating scale (GARS-2), and a behavior rating inventory of executive functioning-preschool version (BRIEF-P) were used to compare the three groups. The results showed that executive functions and behavioral flexibility were more impaired in children with ASD than in children with PDNAS and in TDC. Also, we found that there was no significant difference in the severity of autism symptoms between the children with ASD and the children with PDNAS. Early excessive exposure to digital devices may cause autism-like symptoms in children (PDNAS). Children with PDNAS are different from children with ASD in executive functions and behavioral flexibility. Further research is needed in this area.

## 1. Introduction

Over the last two decades, the prevalence of autism spectrum disorders (ASD) has increased the most compared to other exceptional children groups [1]. Volkmar and Paul [2] reported that since 1992 the prevalence of ASD has increased by more than five hundred percent. Moreover, according to estimates from CDC’s Autism and Developmental Disabilities Monitoring Network, about 1 in 54 children has been identified with ASD [3]. Interestingly, as the prevalence of ASD has increased, the exposure of young children to digital devices (smartphones, tablets, televisions, etc.) has increased in recent years, and in parallel [4,5].

Waldman et al. (2006) investigated whether early childhood television viewing may cause autism, and they concluded that early childhood television viewing is an important trigger for autism [6]. Heffler and Oestreicher [7] proposed a model that explains how digital devices can cause ASD. In this model it is assumed that early excessive screentime in young children, as an inappropriate environmental experience during critical periods of brain development and neuroplasticity, increases the risks of the formation of nonsocial neural circuits. Accordingly, the formation of nonsocial neural circuits may be associated with the emergence of ASD, or at least with ASD-like symptoms [7]. According to Hermawati et al. (2018), early exposure to electronic screens is associated with autistic-like behaviors (e.g., delayed language, short attention span, hyperactivity) in children [8]. It was found by Yurika and Hiroyuki [9] that too much screentime is linked to autistic symptoms in children (i.e., ocular problems, hyperactivity, and language delay). In a recent study, Heffler and Sienko [10] investigated the relationship between early-life social and digital media experiences and autism spectrum disorder–like symptoms. They found that at 12 months of age, prolonged exposure to television or video and shorter caregiver–child interactive play were associated with more significant ASD-like symptoms, but not with autism risk. Moreover, recently Chen and Strodl [11] studied 29,461 preschoolers aged 0 to 3 to investigate how early electronic screen exposure was related to autistic-like behaviors. They found that screen time between the ages of 0 and 3 years was associated with autistic-like behaviors in preschool children and that the association increased with increasing daily screentime. The summary of studies that investigated the relationship between early excessive screentime and ASD symptoms is presented in Table 1.

It appears that when a child is continually exposed to digital devices, the devices replace the child’s dynamic relationship with the social environment. In this lifestyle labelled as “Digital Nanning” [digital nannying] [19], children appear to be at increased risk of losing the opportunity to interact with their mothers or caregivers. Such children appear to be at increased risk of having a poorer social environment for the normal development of the nervous system [19]. Further, “digital nannying” refers to the condition of young children who are exposed to digital devices for more than half of their waking hours [17]. Indeed, digital nannying is a style of care in which digital devices replace the child’s active relationship with his or her environment. In this lifestyle, parents entertain the child with digital devices for various reasons such as being busy with job assignments, lack of skills in communicating with the child, mental health problems such as maternal depression, or the child’s learning from content from the digital devices (such as learning a second language). In this situation, the child, who is in the critical periods of brain development for learning communication and social skills, is deprived of necessary social stimuli. The continuation of these conditions can affect the architecture of the neural networks. So, the formation of nonsocial neural networks may be associated with the emergence of ASD symptoms.

In our previous study [15,21], we used “Post Digital Nanning Autism Syndrome (PDNAS)” to refer to a condition where young children develop subclinical autism symptoms due to early excessive exposure to digital devices (for more than half of their waking hours). We hypothesized it as a new subtype of autistic spectrum disorder, caused by and associated with the lifestyle of the children and early excessive screentime [15,21]. In this case, it is assumed that young children’s autism symptoms are probably causally associated with the early excessive exposure to digital devices. It should be noted that PDNAS is still not an official diagnosis recognized by the American Psychiatric Association or the WHO. So, at the moment, PDNAS should be considered as a provisional label. At this time, because of the similarity of PDNAS to formal ASD, generating a differential diagnosis is impossible. Although in clinical case observations we have found some differences in the symptoms of the two groups of children (such as more visual hyposensitivity and better joint attention in children with PDNAS as compared to the children with ASD), more studies are necessary to study PDNAS. Similarly, Zamfir [22] suggested that early media exposure represents a different type of autism, called “virtual autism.”

Although screen media have many benefits, excessive engagement with them may result in social isolation and deprivation [23,24,25,26]. Evidence suggests that restricted social environments and experiences, as well as early social isolation, can lead to autism symptoms and repetitive behaviors. For example, Rutter et al. (1999 and 2007) found that institutionalized children were more likely to exhibit ‘quasi-autistic’ behavior patterns [27,28]. In another study, Hoksbergen et al. (2005) suggested a similar disorder that was described as a post-institutional autistic syndrome (PIAS) in 16 percent of previously institutionalized infants [29]. In spite of the overlap of symptoms between PIAS and quasi-autism, including social communication difficulties and repetitive behaviors, these researchers described several characteristics that differ from those commonly found in children with well-defined autism spectrum disorders. For instance, while in ASD the female: male ratio is 4:1, in children with PIAS it is 1:1. Further, unlike among children with ASD, among children with PIAS the severity of the autism symptoms appeared to decrease over time [30].

The relationship between early excessive screentime and the onset of ASD symptoms has attracted increased attention. In the present study, we report the findings of a comparison between typically developing children (TDC), young children diagnosed with ASD, and young children with subthreshold autism symptoms with no formal diagnosis of ASD; the children in the latter group had been exposed to screentime for more than half of their waking hours (PDNAS). The aim of the present study was to compare the dimensions of screentime exposure, executive functions, symptoms of ASD, and behavioral flexibility among these three groups. We argue that the present study has the potential to shed further light on the issues of screentime exposure and cognitive performance and social behavior in selected individuals in early childhood.

## 2. Materials and Methods

### 2.1. Participants

This was a comparative cross-sectional study with three different groups of children aged between 22 and 48 months. The first group consisted of children with subthreshold autism symptoms and no formal ASD, though they were exposed to screentime for more than 50% of their waking hours (*n* = 15). These children were referred to the Tehran Autism Center (Tehran University of Medical Sciences, Tehran, Iran) as suspected of having ASD. The second group consisted of 15 gender- and age-matched children clinically diagnosed with ASD. The third group consisted of 15 gender- and age-matched typically developing children (TDC).

ASD specialists with PhDs in clinical psychology evaluated all the children along with an assistant with a Master’s degree in clinical psychology. The exclusion criteria for the subjects were known psychological or genetic syndromes or hearing and visual deficits, as thoroughly assessed by an experienced medical doctor or as taken from medical records.

The inclusion criteria for the children with suspected ASD were: (1) age between 20 and 50 months; (2) ability to undergo a thorough psychological and clinical assessment and to comply with the study conditions; and (3) Parents signed the written informed consent.

The inclusion criteria for the children with ASD were: (1) age between 20 and 50 months; (2) diagnosis of ASD based on the DSM-5 [31] and on a thorough psychological and clinical assessment to comply with the study conditions; and (3) Parents signed the written informed consent.

The inclusion criteria for TDC were: (1) age between 20 and 50 months; (2) ability to undergo a thorough psychological and clinical assessment and to comply with the study conditions; and (3) Parents signed the written informed consent.

### 2.2. Procedure

The study objectives and the anonymous and secure handling of the data were explained to the parents of the children. Thereafter, the parents signed the written informed consent. The parents completed the Gilliam autism rating scale—second edition (GARS-2) [32] (see below); further, the parents reported on the child’s screen time and communication duration (see below). Experienced clinical psychologists performed a thorough standardized testing and behavioral observation of the children. In parallel, experts performed a thorough interview with the parents to gather further information on the children’s behavior. The diagnosis of a child’s ASD followed the diagnostic criteria of the DSM-5 [31] and was made on the basis of a clinical judgment following the interaction with the child, a formal test, and a review of the parents’ reports and records. The duration of the assessment procedure was between 90 and 120 min. The Ethics Committee of the Shahid Beheshti University approved the study (code of ethics: SBU.ICBS 96/1020), which was performed in accordance with the seventh and revised [33] version of the Declaration of Helsinki.

### 2.3. Measures

#### 2.3.1. Sociodemographic Information

Parents reported on their child’s age (months) and gender at birth (male; female). Parents further reported on their own age (years), gender at birth (male; female), number of children, highest educational degree (high school; undergraduate; Master’s degree; PhD/MD), and current employment status (employed; unemployed).

#### 2.3.2. Children’s Lifestyle

To assess the children’s screentime and communication duration, the parents completed a checklist to report their child’s estimated duration of screentime, interaction time, and sleep time. The questionnaire provided the parent-rated duration of the child’s wake time, sleep time, screentime, and interaction time during a given day (unit hours). Screen-time was further refined in terms of foreground and background screentime. Foreground screentime refers to the amount of time a child is in front of a screen to watch programs specifically designed for children. Background screentime refers to the amount of time a child is exposed to screentime of programs not designed for children. We calculated the average hours of the children’s foreground and background screentime, communication time, wake time, and sleep time.

#### 2.3.3. Symptoms of Autism Spectrum Disorders (ASD)

Symptoms of ASD were assessed using two measures. The parents completed the GARS-2 [32] and the modified checklist for autism in toddlers (M-CHAT) [34]. The GARS-2 scale is a well-known tool for diagnosing ASD in children. The GARS-2 consists of 42 items divided into three subscales: stereotyped behaviors, communication, and social interaction. This scale has good sensitivity and specificity for Iranian children [35,36,37,38,39]. Higher sum/mean scores for the subscales and the total scale reflect a higher ASD severity (Cronbach’s alpha = 0.87).

In addition to GARS-2, we used M-CHAT because it is validated for children under three years old. This checklist consists of 23 yes/no items [34]. Since later revised versions of M-CHAT are not standardized for Iranian children (e.g., M-CHAT R/F), we used the earlier version. M-CHAT is a simple screener to assess risk for ASD among toddlers between 16 and 30 months of age. Items are summed up, and a higher sum score reflects a higher risk of ASD (Cronbach’s alpha = 0.77).

#### 2.3.4. Children’s Behavioral Flexibility

To assess a child’s behavioral flexibility, the parents completed the Behavioral Flexibility Rating Scale-Revised (BFRS-R) [40]. This is a parent-report scale that provides a reliable rating of behavioral flexibility. BFRS-R includes 16 items loading on three factors: flexibility toward the environment, flexibility toward objects, and flexibility toward persons. Saniee and Pouretemad [41] reported that BFRS-R has high internal consistency in Iranian children with ASD. The sum of the three factors and the total score are calculated. Higher sum scores reflect a lower behavioral flexibility, or a higher behavioral inflexibility (Cronbach’s alpha = 0.73) [42].

#### 2.3.5. Children’s Executive Functioning

The Behavior Rating Inventory of Executive Functioning—Preschool Version (BRIEF-P) was completed by the parents to assess the children’s executive functioning [43]. This questionnaire is used to assess everyday behaviors in children from 2 to 5.11 years old related to specific domains of executive function (EF). There are 63 items in the BRIEF-P, organized into nine subscales. There are five clinical subscales: working memory, inhibition, emotional control (EC), shifting, and planning/organization. There are also three components: the inhibitory self-control index (ISCI), consisting of the inhibition and EC subscales; the flexibility index (FI), consisting of the shift and EC subscales; and the emergent metacognition index (EMI), consisting of the working memory and plan/organize subscales. Prior studies have reported that the BRIEF-P is valid and reliable [44,45,46]. According to Sadeghi and Pouretemad [16,47], all the subscales of the BRIEF had high internal consistencies in the Iranian children population (Cronbach’s alpha = 0.80 to 0.97). The total score reflects the global executive composite scale (GEC), where a *higher* sum score reflects a *lower* executive function performance, or higher executive function problems.

### 2.4. Statistical Analysis

The data were analyzed by analysis of variance (ANOVA) with SPSS_22_ software [48]. Before using the ANOVA, to evaluate the normality of the data, the Kolmogorov–Smirnov Z (KS-Z) test was used. According to the KS-Z results, all the variables’ scores follow a normal distribution (*p* > 0.05). Moreover, before we performed the ANOVA, we conducted the Levene’s test to ensure that our datasets met the homogeneity of variance assumption. As a result of the Levene’s test, the groups we are comparing have equal variances (*p* > 0.05).

## 3. Results

The behavioral and cognitive differences between young children with PDNAS, young children with ASD, and young TDC were evaluated by one-way ANOVA. The demographic information for the subjects is presented in Table 2.

The one-wat ANOVA results indicated no significant differences between the three groups in age (*F*_(2)_ = 2, *p* = 0.149), sleep duration (*F*_(2)_ = 0.33, *p* = 0.723), or background screentime duration (*F*_(2)_ = 1.16, *p* = 0.324). In contrast, the analyses revealed significant differences among the three groups in foreground screentime duration (*F*_(2)_ = 26.75, *p* = 0.0001) and communication duration (*F*_(2)_ = 8.66, *p* = 0.001). LSD post hoc test results revealed that foreground screentime duration was higher in children with PDNAS than in children with ASD and in TDC, but there was no significant difference between the children with ASD and the TDC. Moreover, the results showed that the communication duration was lower in children with PDNAS than in children with ASD and in TDC, but there was no significant difference between the children with ASD and the TDC.

The analyses revealed no significant difference between the ASD and PDNAS groups in the severity of the children’s ASD symptoms in the M-CHAT, the severity of stereotyped behaviors, or social interactions in the GRAS-2. However, the severity of these symptoms was significantly lower in the TDC than in the children with ASD and PDNAS (Table 3).

The behavioral flexibility problems of the children in the three groups were compared and are reported in Table 4.

The results of the ANOVA showed that the children with ASD have significantly more behavioral flexibility problems than children with PDNAS and TDC. No significant difference was found between the children with PDNAS and the TDC.

We compared the severity of executive function problems in the children with ASD, those with PDNAS, and the TDC, as shown in Table 5.

The results in Table 5 show that the children with ASD had more problems in executive functions than the children with PDNAS and the TDC. No significant difference was found between the children with PDNAS and the TDC.

## 4. Discussion

To the best of our knowledge, this is the first study to differentiate PDNAS from ASD in young children. The behavioral data showed no significant difference in the severity of symptoms of autism between children with ASD and children with PDNAS. In the Materials and Methods section we noted that the PDNAS group had the subthreshold of autism and had not been diagnosed with autism, but these results showed there was no significant difference in severity of autism symptoms between the ASD and PDNAS groups. It is important to clarify that the diagnosis was made by the clinician. So, the clinical evaluation served as the basis of the autism classification. The GARS-2 and M-CHAT scales were filled out by the parents. Contrary to clinical evaluations, these are screening scales with a moderate sensitivity to symptom severity. In this section, we have compared the results of the GARS-2 and M-CHAT screening scales reported by the parents. We found that the symptoms of autism spectrum disorders were significantly lower in the TDC. There was a significant difference among the three groups regarding executive functions and behavioral flexibility. The results showed that executive functions impairment and behavioral inflexibility were significantly lower in the children with PDNAS and in the TDC than in the children with ASD.

One explanation for the lack of significant differences between the children with ASD and the children with PDNAS in the severity of autism symptoms could be their young age. ASD is a developmental disorder and its symptoms become more pronounced with age. Therefore, when children are very young, it may not be easy to differentiate between the signs of the two groups. We also think that the tools used to measure symptom severity and repetitive behaviors were not very sensitive. At a young age, the symptoms of autism mainly include problems with nonverbal social communication skills (such as shared attention, body language, separation anxiety, and eye contact). So, unfortunately these tools are not sensitive enough to compare these symptoms between the two groups. Qualitative and observational studies seem to be better for this comparison. For these reasons, these findings need to be interpreted with caution.

Our findings showed that behavioral flexibility is most affected in children with ASD. The deficit in behavioral flexibility is one of the core symptoms of ASD [49]. This finding is consistent with previous studies that have reported behavioral rigidity in children with ASD [40,49,50]. D’Cruz and Ragozzino [51] have shown that behavioral flexibility is reduced in children with ASD compared with normal children.

Further, our results demonstrated that executive functions were more impaired in children with ASD. This finding is consistent with previous studies that have shown deficits in the executive functions of individuals with ASD [52,53,54,55]. It has been estimated that 41% to 78% of people with ASD display executive dysfunctions [56]. Moreover, it has been proposed that executive dysfunction lies at the heart of ASD [57,58]. There is a large body of evidence that shows that executive dysfunction should be considered a central deficit in persons with ASD [59,60]. Executive dysfunction is exhibited by people with ASD regardless of their performance level and age [61,62]. Overall, this study showed that children with ASD are different from children with PDNAS and from TDC in terms of executive functions and behavioral flexibility.

Executive function and behavioral flexibility, as essential factors in the endophenotype of ASD, were impaired in children with ASD compared to children with PDNAS and with TDC. In recent years, researchers have drawn attention to the early excessive exposure to digital devices as a risk factor and trigger for autism spectrum disorders. Our findings show that children with PDNAS differ from children with ASD in terms of executive functions and behavioral flexibility rather than in terms of severity of autism symptoms. In our opinion, early excessive exposure to digital devices causes autism-like symptoms in young children. We use the term PDNAS for this group of children. Although these children are similar in their symptoms to children who have been formally diagnosed with autism, they are different in terms of cognitive and behavioral characteristics. PDNAS appears to be a new subtype of autism spectrum disorders. We believe that PDNAS is caused by environmental factors such as early excessive exposure to digital devices and social isolation.

We are aware that our study may have some limitations. The first limitation has to do with sampling. Our sample group was small, and this may affect the validity of our findings. The second limitation has to do with measurement. To assess the symptoms, we used the self-report questioner which the parents had to fill in. The third limitation is that this was a cross-sectional study. Longitudinal studies are needed for further clarification; moreover, experimental studies will need to be conducted. Fourth, although our study included children with excessive use of digital devices who were referred to an autism evaluation clinic (PDNAS), it did not include typically developing children in the nonclinical population who excessively used digital devices. A future study should include typically developing children who have excessively used digital devices to determine if there are any differences between them and typical developing children without excessively used digital devices. This can be used to clarify the contribution of excessive use of digital devices to autism symptoms in typically developing children.

## Figures and Tables

**Table 1 jcm-11-06786-t001:** Studies of the relationship between digital device exposure and ASD.

No.	Study	Authors (Year)
1	Early Electronic Screen Exposure and Autistic-Like Behaviors among Preschoolers: The Mediating Role of Caregiver–Child Interaction, Sleep Duration, and Outdoor Activities	Chen and Strodl [11]
2	Association of Early-Life Social and Digital Media Experiences with Development of Autism Spectrum Disorder–Like Symptoms	Heffler and Sienko [10]
3	Early Media Overexposure Syndrome Must Be Suspected in Toddlers Who Display Speech Delay with Autism-Like Symptoms	Dieu-Osika and Bossière [12]
4	Screen time in 36-month-olds at increased likelihood for ASD and ADHD	Hill and Gangi [13]
5	Screen time exposure and severity of autism	Salame and Krayem [14]
6	Parent–child interaction effects on autism symptoms and EEG relative power in young children with excessive screen-time	Sadeghi and Pouretemad [15]
7	Effects of parent–child interaction training on children who are excessively exposed to digital devices: A pilot study	Sadeghi and Pouretemad [16]
8	Behavioral and electrophysiological evidence for parent training in young children with autism symptoms and excessive screen-time	Sadeghi and Pouretemad [17]
9	Screen media and autism spectrum disorder: A systematic literature review	Slobodin and Heffler [18]
10	Digital Nanning and Autism Spectrum Disorder	Pouretemad and Sadeghi [19]
11	Causation model of autism: Audiovisual brain specialization in infancy competes with social brain networks	Heffler and Oestreicher [7]
12	Early electronic screen exposure and autistic-like symptoms	Hermawati and Rahmadi [8]
13	Attachment Disorder and Early Media Exposure: Neurobehavioral symptoms mimicking autism spectrum disorder.	Numata-Uematsu and Yokoyama [9]
14	Does television cause autism?	Waldman and Nicholson [6]
15	The role of digital nursing in shaping of autistic disorders symptoms	Pouretemad [20]

**Table 2 jcm-11-06786-t002:** Sociodemographic characterization of the families and children.

Children’s age and lifestyle characteristics	**Group**	**Sex**	**N**	**Age (Month)**	**CFSTD (h)**	**CBSTD (h)**	**CCD (h)**	**CSD (h)**
	PDNAS	Male	11	24.80	7.60	3.80	1.77	11.42
		Female	4					
	ASD	Male	11	27.33	2.66	4.10	4.34	11.16
		Female	4					
	TDC	Male	5	22.15	1.30	2.50	6.24	11.37
		Female	10					
**Sociodemographic Characterization of the Families**								
Sociodemographic characterization of the families	**Variables**		**PDNAS Group**		**ASD Group**		**TDC Group**	
			Mother	Father	Mother	Father	Mother	Father
			34.20	38.87	32.60	35.40	31.69	34.69
	Number of children in the family	One child	9		10		11	
		Two children	6		4		4	
		Three children	0		1		0	
	Education	High School	2	4	2	6	2	3
		Undergraduate	10	4	10	4	1	4
		Master	3	4	2	5	9	4
		Doctoral	0	3	1	0	3	4
	Economically active	Work	1	15	2	14	6	13
		Do not Work	14	0	13	1	9	2

Abbreviations: PDNAS, Post–Digital Nannying Autism Syndrome; ASD, autism spectrum disorders; TDC, typically developing children; CFSTD, children’s foreground screentime duration; CBSTD, children’s background screentime duration; CCD, children’s communication duration; CSD, children’s sleep duration; h, hour.

**Table 3 jcm-11-06786-t003:** Comparison of the severity of autism symptoms in two groups.

Variables		Group	M	SD	F	p	LSD Pairwise Comparison
M-CHAT		PDNAS	33.33	4.34	18.35	0.0001	PDNAS > TDC **** ASD > TDC ****
		ASD	31.60	4.70			
		TDC	24.84	1.46			
GARS-2	Stereotypical behaviors	PDNAS	13.53	8.20	3.55	0.038	PDNAS > TDC * ASD > TDC *
		ASD	13.87	7.31			
		TDC	7.69	5.64			
	Social interactions	PDNAS	17.27	8.33	9.80	0.0001	PDNAS > TDC **** ASD > TDC ****
		ASD	16.93	7.93			
		TDC	6	5.99			
	Total scores	PDNAS	30.80	14.34	8.09	0.001	PDNAS > TDC *** ASD > TDC ***
		ASD	30.80	14.68			
		TDC	13.69	7.90			

Abbreviations: PDNAS, Post–Digital Nannying Autism Syndrome; ASD, autism spectrum disorders; TDC, typically developing children; M, Mean; SD, Standard Deviation; LSD, Least Significant Difference; *, *p* < 0.05; ***, *p* < 0.001; ****, *p* < 0.0001.

**Table 4 jcm-11-06786-t004:** Comparison of the severity of behavioral flexibility problems in three groups.

Variables	Group	M	SD	F	p	LSD Pairwise Comparison
flexibility towards the environment	PDNAS	2.40	2.64	5.51	0.008	PDNAS < ASD * TDC < ASD **
	ASD	4.20	2.30			
	TDC	1.38	1.76			
flexibility towards objects	PDNAS	4.73	4.03	5.57	0.007	PDNAS < ASD ** TDC < ASD **
	ASD	10.47	7.64			
	TDC	4.31	4			
flexibility towards persons	PDNAS	2.93	1.87	3.89	0.029	PDNAS < ASD * TDC < ASD *
	ASD	4.80	2.73			
	TDC	2.54	2.29			
Total Score	PDNAS	9.47	6.85	5.82	0.006	PDNAS < ASD * TDC < ASD **
	ASD	17.07	9.78			
	TDC	7.36	8.88			

Abbreviations: PDNAS, Post–Digital Nannying Autism Syndrome; ASD, autism spectrum disorders; TDC, typically developing children; M, Mean; SD, Standard Deviation; LSD, Least Significant Difference; *, *p* < 0.05; **, *p* < 0.01.

**Table 5 jcm-11-06786-t005:** Comparison of executive function problems in three groups.

Variable	Group	M	SD	F	p	LSD Pairwise Comparison
Inhibition	PDNAS	11.80	6.25	16.10	0.0001	ASD > PDNAS **** ASD > TDC ****
	ASD	27.67	11.88			
	TDC	9.77	8.86			
Shifting	PDNAS	9	6.20	8.72	0.001	ASD > PDNAS ** ASD > TDC ****
	ASD	16.53	8.02			
	TDC	6.84	4.70			
Emotional control	PDNAS	8.60	4.76	11.80	0.0001	ASD > PDNAS ** ASD > TDC ****
	ASD	16.40	7.16			
	TDC	6.23	5.34			
Working memory	PDNAS	9.87	6.97	15.23	0.0001	ASD > PDNAS **** ASD > TDC ****
	ASD	26.73	13.36			
	TDC	8.54	7.94			
Planning/Organization	PDNAS	8.60	4.76	11.80	0.0001	ASD > PDNAS *** ASD > TDC ****
	ASD	16.40	7.16			
	TDC	6.23	5.34			
Self-Control Index (ISCI)	PDNAS	20.40	10.34	15.24	0.0001	ASD > PDNAS **** ASD > TDC ****
	ASD	44.06	18.64			
	TDC	16	13.80			
Flexibility Index (FI)	PDNAS	17.60	9.79	11.16	0.0001	ASD > PDNAS *** ASD > TDC ****
	ASD	32.93	14.84			
	TDC	13.08	9.79			
Emergent Metacognition Index (EMI)	PDNAS	18.46	10.69	15.34	0.0001	ASD > PDNAS **** ASD > TDC ****
	ASD	43.13	20			
	TDC	14.77	12.21			
Global Executive Composite (GEC)	PDNAS	44	22.08	15.79	0.0001	ASD > PDNAS **** ASD > TDC ****
	ASD	102	45.66			
	TDC	37.46	29.81			

Abbreviations: PDNAS, Post–Digital Nannying Autism Syndrome; ASD, autism spectrum disorders; TDC, typically developing children; M, Mean; SD, Standard Deviation; LSD, Least Significant Difference; **, *p* < 0.01; ***, *p* < 0.001; ****, *p* < 0.0001.

## Data Availability

The data used to support the findings of this study are included in the article.

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
