# Peer review of "Differentiating Post–Digital Nannying Autism Syndrome from Autism Spectrum Disorders in Young Children: A Comparative Cross-Sectional Study"

_jcm, 2022, doi:10.3390/jcm11226786_

Round 1
Reviewer 1 Report
Dear Authors:
The article entitled "Differentiation of Post-Digital-Nanning Autism Syndrome from 2 Autism Spectrum Disorder in Young Children: A cross-sectional comparative 3 study", presents an adequate structure and the language used is easy to read. However, some changes and improvements are needed.
At a formal level, it is recommended to use the concept of "Autistic Spectrum Disorders" in the plural to indicate that it is a non-specific set of disorders (some syndromic, others idiopathic), characterised by sharing two symptomatological dimensions, social communication disorders or deficits and repetitive behaviours and restricted interests. On the other hand, a thorough revision of the text is needed. In some sections there are sentences that suggest that it is a working draft. For example in lines 224-225 it is stated: "I would strongly suggest that the correlation coefficients between the dimensions assessed, both for the whole group and separately for three groups, should also be reported".
In terms of content, the concept of "Digital Nanning" requires further reflection and definition. To speak of "Digital Nanning" as the condition that young children have been exposed to digital devices for more than half of their waking hours is poor and insufficient and relates to more permissive (laissez-faire) parenting and family lifestyles. Leaving or abandoning children in front of screens for hours on end may involve disruptions in the family or personal lives of the parenting partner.
In addition, it is necessary to report whether Post Digital Autism Syndrome (PDNAS) is a disorder recognised by the WHO, the American Psychiatric Association or another organisation. If it is not, it should be indicated as a provisional label, unless it is already used by other researchers outside the research team.
For the "measures" section, a lifestyle checklist is mentioned (this reviewer has not had access to the supplementary material), and if it is a standard checklist, it would be necessary to cite the authors, and if it is an ad hoc checklist, it would be necessary to provide data on its test-retest stability and inter-observer validity.
For the diagnosis of ASD the MCHAT is said to be used (Robins et al., 2001), however, there are later revised versions (Robins et al 2013) and even improved versions with follow-up interviews (Mchat R/F). The rationale for using the 2001 version needs to be justified.
In the procedure section it is stated that: "Experienced clinical psychologists conducted a comprehensive standardised test, and behavioural observation of the children. In parallel, the experts conducted an in-depth interview with the children's parents to gather more information about their behaviour". Conventionally, the ADOS-2 (Lord et al 2012, Autism Diagnostic Observation Schedule) is used for the behavioural observation and the ADI-R (Le Couteur et al 2003, Autism Diagnostic Interview-Revised) for the parent interview. were these instruments used? And if not, which ones were used?
In the statistical analysis section, it is indicated that the Kolmogorov-Smirnov test was used to assess the normality of the data, but the result is not reflected in any table.
Since the size of the groups is very small, to determine the appropriateness of using ANOVA it is necessary to provide information on distribution fit, kurtosis and symmetry. Otherwise, it is advisable to use an alternative non-parametric analysis, e.g. the Kruskal-Wallis test. Or adequately justify the choice of test.
Check the use of acronyms in the tables. For example, in table 3 the acronym LSD appears but the meaning is not specified. On the other hand, the term Eta squared appears as an indicator of the power of the effect but the value is not specified. The a posteriori contrast requires an explanation and perhaps a separate table.
Author Response
We thank Reviewer #1 for their valuable comments, which helped us to improve the quality of the revised manuscript. Please find attached the point-by-point-response and the revised ms.
Thank you again for all your kind efforts.

Reviewer 2 Report
In this study, the authors aim to differentiate children with Post Digital-Nanning Autism Syndrome (PDNAS) from children with autism spectrum disorder (ASD) and typically developing children (TDC), both behaviorally and cognitively. This research is a comparative cross-sectional study. The study included three groups of children. The first group consisted of 15 young children with subthreshold autism symptoms. They had not received a diagnosis of ASD and had been exposed to digital devices for more than half of their waking time. The second group consisted of 15 young children with ASD, and the third group consisted of 15 young TDC. Lifestyle checklist, modified checklist for autism in toddlers (M-CHAT), behavioral flexibility rating scale-revised (BFRS-R), Gilliam autism rating scale (GARS-2), and behavior rating inventory of executive functioning-preschool version (BRIEF-P) were used to compare the three groups. Results showed that executive functions and behavioral flexibility were more impaired in children with ASD than in children with PDNAS and TDC. Also, we found that there was no significant difference in the severity of autism symptoms between children with ASD and children with PDNAS. Early excessive exposure to digital devices may cause autism-like symptoms in children (PDNAS). Children with PDNAS are different from children with ASD in executive functions and behavioral flexibility. Experimental studies are needed in this area.
Strengthens:
This study provides a novel sight to study ASD, which is interesting. The analysis and presentation is clear and good.
Weakness:
1. The three groups include: ASD group; Typically developing group; the group with subthreshold of ASD (had not received a diagnosis of ASD and had been exposed to digital devices. In the results, the authors found that the children with ASD showed more impaired executive functions and behavioral flexibility than PDNAS and TDC. But there is no significant difference in the severity of autism symptoms between ASD and PDNAS groups.
PDNAS group are with subthreshold of ASD and had not received a diagnosis of ASD, why is there no significant difference in severity of autism symptoms between ASD and PDNAS groups?
2. In this study, the authors only include children with autistic symptoms who had been exposed to digital devices, I am just wondering if typically developing children who have been exposed to digital devices show any difference from typically developing children who weren’t exposed to digital devices? Some discussion may need to be included in the manuscript.
Author Response
We thank Reviewer #2 for their valuable comments, which helped us to improve the quality of the revised manuscript. Please find attached the point-by-point-response and the revised ms.
Thank you again for all your kind efforts.

Round 2
Reviewer 2 Report
The authors have fully addressed my concerns about the manuscript.